# Seven Weeks of Jump Training with Superimposed Whole-Body Electromyostimulation Does Not Affect the Physiological and Cellular Parameters of Endurance Performance in Amateur Soccer Players

**DOI:** 10.3390/ijerph17031123

**Published:** 2020-02-10

**Authors:** Nicolas Wirtz, André Filipovic, Sebastian Gehlert, Markus de Marées, Thorsten Schiffer, Wilhelm Bloch, Lars Donath

**Affiliations:** 1Institute of Training Science and Sport Informatics, Department of Intervention Research in Exercise Training, German Sport University Cologne, 50933 Cologne, Germany; l.donath@dshs-koeln.de; 2Institute of Cardiology and Sports Medicine, Department of Molecular and Cellular Sports Medicine, German Sport University Cologne, 50933 Cologne, Germany; andre.filipovic@gmx.net (A.F.); gehlert@dshs-koeln.de (S.G.); w.bloch@dshs-koeln.de (W.B.); 3Institute of Sport Science, University of Hildesheim, 50933 Hildesheim, Germany; 4Section of Sports Medicine and Sports Nutrition, Faculty of Sports Science, Ruhr University of Bochum, 44801 Bochum, Germany; Markus.deMarees@ruhr-uni-bochum.de; 5Outpatient Clinic for Sports Traumatology and Public Health Consultation, German Sport University Cologne, 50933 Cologne, Germany; t.schiffer@dshs-koeln.de

**Keywords:** electrostimulation, soccer, lactate, VO_2_peak, monocarboxylate transporter

## Abstract

Intramuscular density of monocarboxylate-transporter (MCT) could affect the ability to perform high amounts of fast and explosive actions during a soccer game. MCTs have been proven to be essential for lactate shuttling and pH regulation during exercise and can undergo notable adaptational changes depending on training. The aim of this study was to evaluate the occurrence and direction of potential effects of a 7-weeks training period of jumps with superimposed whole-body electromyostimulation on soccer relevant performance surrogates and MCT density in soccer players. For this purpose, 30 amateur soccer players were randomly assigned to three groups. One group performed dynamic whole-body strength training including 3 x 10 squat jumps with WB-EMS (EG, *n* = 10) twice a week in addition to their daily soccer training routine. A jump training group (TG, *n* = 10) performed the same training routine without EMS, whereas a control group (CG, *n* = 8) merely performed their daily soccer routine. 2 (Time: pre vs. post) x 3 (group: EG, TG, CG) repeated measures analyses of variance (rANOVA) revealed neither a significant time, group nor interaction effect for VO_2_peak, Total Time to Exhaustion and La_max_ as well as MCT-1 density. Due to a lack of task-specificity of the underlying training stimuli, we conclude that seven weeks of WB-EMS superimposed to jump exercise twice a week does not relevantly influence aerobic performance or MCT density.

## 1. Introduction

The physical demands of soccer players have increased notably within the last 10 to 20 years due to modern game tactics and their variability. For example, the ability of a team to successfully play high pressing mainly depends on the physical characteristics of the players. The distances covered in the higher intensities and the number of quick and explosive actions such as accelerations, turns, and jumps have increased within the recent years [1,2,3]. A player’s capacity to perform numerous of those actions with a highly intense load is considered crucial in modern soccer. This ability relies on (1) adequate intra- and intermuscular coordination of soccer-specific movements and (2) metabolism that ensures proper energy delivery [4]. Both has been shown to be affected by electromyostimulation (EMS) training [5,6].

Jumps with superimposed Whole-Body EMS (WB-EMS) in addition to soccer training sessions can be effective for improving accelerations, turns, jumps, and kicking velocity [5]. WB-EMS potentially supports the athlete achieving higher power outputs and faster sport-specific movement velocities using resistance training [7] by increased firing rates and synchronization of motor units, resulting in a more pronounced activation of fast-twitch fibers at relatively low force levels [8]. Previous studies showed that local EMS is beneficially affecting muscle metabolism and can elevate energy expenditure and carbohydrate oxidation to a higher degree than voluntary contraction only [9,10,11]. Moreover, WB-EMS seem to stimulate anaerobic glycolysis for energy production with higher lactate accumulation [12,13]. The beneficial effects of EMS on transportation of lactate have to be taken into account as lactate shuttling via monocarboxylate transporters (MCTs) has been shown to improve high-intensity intermittent exercise performance [14,15].

MCTs are considered essential for lactate shuttling and pH regulation during exercise and can undergo notable adaptational changes depending on physical activity levels [16,17]. Due to a 1:1 ratio of lactate and H+ being transported by MCTs, an increase in the two isoforms MCT-1 and MCT-4 in skeletal muscle reduce the intracellular pH perturbations [18]. In line with this, studies revealed that the density of MCT-1 and MCT-4 proteins in muscle is elevated after a macrocycle of endurance training [19,20,21]. However, some training studies did not find relevant increases in MCT-4 density [22,23,24]. It has been assumed that MCT-1 production is more sensitive to physical stress than MCT-4. Since the biochemical characteristics of MCT-1 favors lactate uptake, it has been suggested that erythrocytes provide a lactate storage compartment in situations of physical exercise, thereby reducing the exercise-induced increase in plasma lactate concentration [25].

Interestingly, Fransson et al. [26] showed remarkable changes in MCT-4 protein expression after 4 weeks of soccer specific training regimes like speed endurance (+30%) and small sided games (+61%) in well-trained soccer players. An increase in MCT-1 and MCT-4 density in skeletal muscle after 6 weeks of strength training was however merely reported by Juel et al. [27]. No available study investigated the effects of WB-EMS on relevant endurance capacities like VO_2max_ and MCT-1 and MCT-4 in soccer players. Against this background, the aim of our 3-armed randomized controlled trial was to elucidate whether WB-EBS supplemented to a traditional soccer training routine can improve endurance capacities indices and MCT density of soccer players. Our primary hypothesis was that a training program of jumps with superimposed EMS may pronouncedly stimulate MCT-1 and MCT-4 density. Our secondary hypothesis was that endurance performance surrogates will not be affected by the training program, because of the relatively low additional training volume and the subject´s high overall training status.

## 2. Material and Methods

### 2.1. Participants

Only healthy field soccer players were included which means no cardiovascular or metabolic diseases and no preinjury in the tested muscle groups. Participants needed to compete on a regional level for the last 3 years and train 2–4 session per week with strength and conditioning training contents and play one soccer match per week. In a randomized control trial twenty-eight soccer players from 10 different teams were assigned to three different groups. Control group was assigned based on preferences and availability, whereas both intervention arms have been assigned based on coin toss. The EMS group (EG, *n* = 10) performed jumps with superimposed WB-EMS twice a week accompanied by 3 × 10 squat jumps in addition to the daily soccer routine over a period of 7 weeks that is a sufficient intervention period with WB-EMS to improve strength abilities [5,28,29]. To differentiate between the effects caused by EMS and by the squat jumps and soccer training respectively, two control groups were included. A jump training group (TG, *n* = 10) performed the same number of squat jumps without EMS on the same days as the EG and a control group (CG, *n* = 8) that only performed the daily soccer routine. All subjects were non-smokers. Basal anthropometric parameters of the subjects were presented in Table 1.

This study was carried out in accordance with the recommendations of the “Ethics Committee of the German Sports University Cologne”. All subjects gave written informed consent in accordance with the Declaration of Helsinki. The protocol was approved by the “Ethics Committee of the German Sports University Cologne” (06–02–2014).

### 2.2. Daily Soccer Routine

The participants performed 3.2 ± 1.0 soccer training sessions per week and competed once a week in the championships. The standard training sessions lasted approximately 90 min including technical skill activities, offensive and defensive tactics, athletic components with various intensities, small-sided game plays and continuous play. In a normal training week during season with a match on Sunday training was scheduled on Tuesdays, Wednesdays (optional), Thursdays and Fridays. Number of training sessions and the training days varied according to the game schedule playing Sunday-Sunday or Sunday-Saturday. The number of training sessions and the total training minutes were documented. The training load was measured according to the training time spent in defined heart rate zones during soccer training or match via Polar Team-2 Software (Polar Electro, Büttelborn, Germany) (see Table 1). The training load [arbitrary units] provided by the Polar-Software aims to determine internal training load based on background variables (sex, training history, metabolic thresholds, and maximal oxygen consumption [VO_2_max]) and parameters measured during training sessions (exercise mode, and energy expenditure) (c.f. [30]). The heart rate zones (100–90%, 89–80%, 79–70%, 69–60%, 59–50%) were defined according to the individual maximum heart rate measured in the maximal ramp test (see endurance test).

The players were asked to maintain their usual food intake und hydration according to the recommendations for soccer players [31] and no nutrition supplementation was used. Additional strength training was not allowed during the study.

All players had a constant training volume during the first half of the season (July till December) and were in a well-trained condition with a relative VO_2_peak of 54.2 ± 5.9 mL/kg·min^−1^. All players regularly conducted strength training during first half of the season and had overall experience in strength training of 5.4 ± 3.9 years. The intervention period started after the three weeks mid-season break from end of December till mid of January. During these three weeks the training load was relatively low (moderate endurance training twice per week) in order to maintain fitness level and not negatively affect Baseline testing.

### 2.3. WB-EMS Application and Protocol

In order to obtain a rest interval of 48 h between the two sessions and the championship game on Sunday WB-EMS training was conducted on Tuesdays and Friday. All subjects abstained from alcohol consumption for 24 h prior to and during the training intervention. The EMS Training was conducted with a WB-EMS-system by Miha Bodytec (Augsburg, Germany). WB-EMS was applied with an electrode vest to the upper body with integrated bilaterally two paired surface electrodes for the chest (15 × 5 cm), upper and lower back (14 × 11 cm), latissimus (14 × 9 cm), and the abdominals (23 × 10 cm) and with a belt system to the lower body including the muscles of the glutes (13 × 10 cm), thighs (44 × 4 cm) and calves (27 × 4 cm). Biphasic rectangular wave pulsed currents (80 Hz) were used with an impulse width of 350 µs [5]. The stimulation intensity (mA) was determined and set separately for each muscle group (0–120 mA) by using a Borg Rating of Perceived Exertion [32]. The training intensity was defined for each player in a familiarization session two weeks before and set at a sub-maximal level that still assures a clean dynamic jump movement (RPE 16–19 “hard to very hard”) and was saved on a personalized chip card. The EG performed 3 × 10 maximal squat jumps with a set pause of 60 s (no currents) per session. Every impulse for a single jump lasted for 4 s (range of motion: 2 s eccentric from standing position to an knee angle of 90°–1 s isometric–0.1 s explosive concentric–1 s landing and stabilisation) followed by a rest period of 10 s (duty cycle approx. 28%). This results in an overall time of 8.5 min per session an effective stimulation time of 2 min per session. The players started with a 2–3 min standardized warm-up with movement preparations including squats, skipping and jumps in different variations (squat jumps, jumps out of skipping or double jumps) at a light to moderate stimulation intensity. The players were told to slowly increase the intensity every few impulses. The training started when the players reached the defined training intensity that was saved on the chip card from the last session according to the RPE 16–19 (“hard to very hard”). The stimulation intensity was constantly increased individually every week (Tuesdays) controlled by the coaches in order to maintain a high stimulation intensity. The intensity was increased after the warm-up during the first and the second set of 10 squat jumps starting from calves up to the chest electrodes. The TG conducted the same standardized warm-up and performed the same amount of jumps with identical interval and conduction twice per week without EMS. The CG only performed the 2–4 soccer training session plus one match per week.

### 2.4. Experimental Protocol

#### 2.4.1. Endurance Test and Assessment of Anthropometrics

For determination of the endurance parameters spirometry was performed on a WOODWAY treadmill (Woodway GmbH, Weil am Rhein, Germany) one week before (Baseline) and after the 7-weeks intervention period (Post-test) (Figure 1). Furthermore, bodymass and body composition were determined via bioelectrical impendence analysis (TANITA corp., Tokyo, Japan). Endurance tests were conducted three days after the soccer match to assure adequate recovery and not negatively influence performance. Respiratory gases were analyzed via the ZAN600-System and ZAN-Software GPI 3.xx (ZAN Austria e.U., Steyr-Dietach, Austria), using standard algorithms with dynamic account for the time delay between the gas consumption and volume signal. To calibrate the device according to the manufacturer´s guidelines, a gas mixture consisting of 5% CO_2_, 16% O_2_, and rest nitrogen was used (Praxair Deutschland GmbH, Düsseldorf, Germany). To measure the maximum oxygen uptake (VO_2_peak), the subjects performed an incremental ramp test [33]. Thereby, the players performed a warmup at moderate speed (3 m·s^−1^) with 1% incline for 3 min. In the last 30 s the incline was increased to 2.5%. Subsequently, running speed was then increased every 30 sec by 0.3 m/s until subjective exhaustion was reported. Heart rate was documented in the last 10s of a ramp stage. The VO_2_peak was determined as average maximum oxygen uptake of the first 20 s after ending the test. Additionally, maximum heart rate, time to exertion (TTE) and maximum lactate concentration (La_max_) was recorded. 27 players completed the two endurance diagnostics. One player of the TG was removed from the study due to an ankle joint injury prior to post testing.

#### 2.4.2. Muscle Biopsies and Tissue Treatment.

Muscle biopsies via Bergström method [34] were taken from each player two weeks before (Baseline) and in the week after the last training intervention (Post-test). All biopsies were obtained under local anaesthetic from the middle portion of the vastus lateralis between the lateral part of the patella and spina iliaca anterior superior 2.5 cm below the fascia. The muscle samples were freed from blood and non-muscular material and embedded in tissue freezing medium (TISSUE TEK, Sakura, Zoeterwoude, The Netherlands). Samples were frozen in liquid nitrogen-cooled isopentane and stored at −80 °C for further analysis. The distance between the Baseline and Post-test incision was approx. 2.5 cm.

### 2.5. Immunohistochemistry

Muscle samples from 26 subjects were used for histology. 7 μm cross-sectional slices were obtained from the frozen muscle tissue using a cryo-microtome Leica CM 3050 C (Leica Microsystems, Nußbach, Germany) and placed on Polysine^™^ microscope slides (VWR International, Leuven, Belgium) [35]. Sections were fixed for 5 min in −20 °C pre-cooled acetone and air dried for 10 min at room temperature (RT), before blocking for one hour at RT with TBS (tris buffered saline, 150 mM NaCl, 10 mM Tris-HCl, pH 7.6) containing 5% BSA (bovine serum albumin). After blocking, sections were incubated overnight (4 °C) with primary antibody for MCT-1 (ab3538P; 1:500; Merck Millipore, Burlington, MA, USA) and MCT-4 (sc-376140; 1:400; Santa Cruz Biotechnology, Dallas, TX, USA), diluted in 0.8% BSA. To confirm antibody specificity, control sections were incubated in TBS containing 0.8% BSA but without primary antibody. After incubation, sections were washed 5 times short and twice for 10 min with TBS and incubated for one hour with biotinylated goat anti-rabbit secondary antibody for MCT-1 (VECTOR Laboratories, Burlingame, CA, USA), diluted 1:500 in TBS and goat anti-mouse for MCT-4 (VECTOR Laboratories), diluted 1:400 in TBS, at RT.

After that, sections were washed again 5 times for 30 s and twice for 10 min before incubation with fluorescent Alexa 488 secondary antibodies (Life Technologies, Carlsbad, CA, USA); diluted 1:500 in TBS for an hour. Afterwards sections were blocked with 5% BSA (TBS-Tween) for 30 min. Slides were then incubated overnight at 4 °C with A4.951primary antibodies (A4951; type-I myosin heavy chain; Developmental Studies Hybridoma Bank, Iowa City, IA, USA) diluted 1:200 in 0.8% BSA.

On the third day, sections were washed 5 times short and twice for 10 min with TBS before incubated again with secondary antibodies and fluorescent Alexa 555 (red) diluted 1:500 in TBS for an hour at RT. After washing again the samples, fixed on microscope glass slides, were embedded with aqualpolymount and stored at RT.

### 2.6. Data Analysis

The analysis of immunofluorescence stained myofibers were conducted with a confocal laser scanning microscope (LSM 510, Zeiss, Jena, Germany) at 63X fold magnification. For the analysis of MCT density in sarcoplasm and myofiber membranes, two laser channels 543 nm (for Alexa 555) and 488 nm (for Alexa 488) were used.

Two separate line-scans were conducted per measurement of membrane and sarcoplasmic areas of single myofibers to determine the staining intensity for MCT 1 and MCT 4 and the means was used for analysis (Figure 2). 1000 pixels were standardized analyzed per line scan along the membrane and the sarcoplasm. MCT density was then calculated as the mean staining intensity of all pixels along each line scan. For the analysis of type I fibers, only the green channel (Alexa 488) was used for analysis and the red channel (Alex 555) was used for fiber type determination. Laser intensity was standardized for each subject without changing throughout the analysis.

### 2.7. Statistical Analysis

To determine the effect of the training interventions on endurance parameters, MCT-1 and MCT-4, separate 2 (time: pre vs. post) × 3 (group: EG, TG, CG) mixed ANOVA with repeated measures were conducted. ANOVA assumption of homogenous variances was tested using Maulchy-test of Sphericity. A Greenhouse-Geisser correction was used when a violation of Mauchly´s test was observed. To estimate overall time and interaction effect sizes, partial eta squared (η^2^_p_) was computed with η^2^_p_ ≥ 0.01 indicating small, ≥0.059 medium and ≥0.138 large effects [36]. If 2 × 3 mixed ANOVA revealed a time*group interaction effect on any variable, this effect was further investigated using Bonferroni post hoc tests for pairwise comparison. For all inferential statistical analyses, significance was defined as a p-value less than 0.05. All descriptive and inferential statistical analyses were conducted using SPSS 25^®^ (IBM^®^, Armonk, NY, USA). Results were presented as means and standard deviations (SDs). Figures were created with Prism 6 (GraphPad Software Inc., La Jolla, CA, USA).

## 3. Results

### 3.1. Training Load

No significant differences were observed between the groups in the total number of training sessions (EG 23.9 ± 7.8; TG 25.9 ± 6.6; CG 18.1 ± 5.6 sessions), training minutes (EG 2103 ± 630; TG 1812 ± 919; CG 1437 ± 381Min), and the total recorded training load via Polar Team-2 software (Table 1). All subjects of TG and EG had a compliance of 100% (14 training sessions) for jump training and WB-EMS sessions, respectively.

### 3.2. Endurance Parameters

2 × 3 (time × group) ANOVA of repeated measures revealed no significant time, group or interaction effect for relative VO_2_peak, TTE and La_max_. No group differences were observed at Baseline or Posttest in none of the analyzed parameters (Figure 3).

### 3.3. MCT-4

#### 3.3.1. Type-I Fibers

The 2 × 3 (time × group) repeated measures ANOVA revealed no significant time (*p* = 0.119, η^2^_p_ = 0.102), group or intervention effect (*p* = 0.165, η^2^_p_ = 0.145) for the MCT-4 density in the membrane of type-I muscle fibers. Regarding cytoplasm density of the MCT-4, a large significant effect over time (*p* = 0.009, η^2^_p_ = 0.26) was shown. No group*time effect (*p* = 0.318, η^2^_p_ = 0.095) was however observed. Subsequent post-hoc analysis showed a significant decrease in MCT-4 density after 7 weeks for TG only (*p* = 0.005). No group differences were detected at Baseline and Posttest for MCT-4 density in membrane and cytoplasm in type-I fibers (Figure 4).

#### 3.3.2. Type-II Fibers

With respect to the membrane density of the MCT-4, no time effect (*p* = 0.172, η^2^_p_ = 0.079) or group*time interaction (*p* = 0.315, η^2^_p_ = 0.096) of type-II fibers was shown. For the cytoplasm density of the MCT-4 a large significant main effect for the factor time (*p* = 0.001, η^2^_p_ = 0.382) was observed in type-II fibers. No group*time interaction effect (*p* = 0.333, η^2^_p_ = 0.091) was observed. Subsequent post-hoc analysis showed a significant decrease in MCT-4 distribution only for TG (*p* = 0.004). No differences were shown between the groups at Baseline or Posttest for MCT-4 density in the membrane and cytoplasm of the type-II fibers (Figure 4).

### 3.4. MCT-1

#### 3.4.1. Type-I Fibers

The 2 × 3 (time × group) repeated measure ANOVA showed no significant effect over time (*p* = 0.230, η^2^_p_ = 0.065) as well as no significant group*time interaction effect (*p* = 0.045, η^2^_p_ = 0.246) for MCT-1 density in the membrane. Post-hoc analysis showed a significant decrease in density for the TG (*p* = 0.032). For the cytoplasm density of the MCT-1 no main effects over time (*p* = 0.114, η^2^_p_ = 0.110) or group*time (*p* = 0.416, η^2^_p_ = 0.077) were found. No group differences were detected at Baseline and Posttest for MCT-1 density in the membrane and cytoplasm of the type-I fibers (Figure 5).

#### 3.4.2. Type-II Fibers

The 2 × 3 ANOVA revealed a large significant time effect for cytoplasm MCT-1 (*p* = 0.009, η^2^_p_ = 0.269) but no group × time interaction effect (*p* = 0.933, η^2^_p_ = 0.006). However, no significant alternations in the cytoplasm were found in the three different groups over time. For membrane density of the MCT-1 neither a time (*p* = 0.104, η^2^_p_ = 0.115) nor an interaction effect (*p* = 0.480, η^2^_p_ = 0.065) was observed in type-II fibers. Group comparison revealed no differences between the three groups at baseline and post-testing for MCT-1 density in the membrane and cytoplasm of the type-II fibers (Figure 5).

## 4. Discussion

The main finding of this intervention is that 7 weeks of a dynamic WB-EMS program (2 sessions per week) in addition to the regular soccer training does not relevantly influence endurance performance indices as well as MCT-1 or MCT-4 density in the muscle. We surprisingly observed that MCT-4 density in the cytoplasm and MCT-1 density in the membrane of type I muscle fibers notably decreased in TG, the group that completed jumps without EMS. Participants of all three groups (EG, TG, CG) did their weekly soccer sessions since years and training volume and training intensity was not changed during the intervention period. Thus, the results could be explained by the high overall training status of the subjects. Due to the documented effects of strength training on runner´s performance [37] and effects of EMS application on runner´s VO_2_max [6], we analysed effects on some endurance parameter for EG. Indeed, WB-EMS intervention with dynamic exercises and training status of the subjects (VO_2_max: 53 mL/min/kg) were similar to the study of Amaro-Gahete et al. [6]. However, the differential results might be attributed to differences in current frequency (12–90 vs. 85 Hz), higher time under tension (6 vs. 2 min) and higher intensities of exercises with superimposed EMS (strength and interval exercise vs. jumps) in the cited study. With regard to the training status of the subjects and the general high metabolic demand in soccer games and -training, the EMS stimulus could have been too low for further adaptations in endurance capacities. Consequently, no changes were observed in any parameter obtained during incremental treadmill running test. With respect to the results of Amaro-Gahete and coworkers and in order to improve endurance parameter, it might be promising to adjust exercise to higher training intensities, e.g., by shortening rest intervals of jumps or include other exercises within high intensity intervals. The authors provided recommendations for an undulating modulation of current adjustments [38], but without physiological explanation or reasoning. There are no studies available that support these results and we found only one study that applied EMS during endurance training. In this regard, Mathes and coworkers showed that, although metabolic stimuli and markers of muscle damage were higher in cycling with superimposed EMS compared to cycling without EMS, improvements of endurance performance and capacity were not significantly different between both training methods [39].

The disposed EMS-protocol concurrently to soccer training enhanced strength and myofiber adaptations [40]. Furthermore it revealed to be effective for accelerations, direction changes, vertical jumping ability and kicking velocity in elite soccer players [5]. Training design was identical within the present study. Improving such surrogate parameters of aerobic or anaerobic endurance capacities seems also promising to improve indices of soccer performance. The ability to perform sprints with high intensity bouts is influenced by anaerobic capacity. The ability to do that repeatedly critically depends on the aerobic metabolism. Both metabolic pathways are inter-linked with each other. However, for the recommendation of WB-EMS, it would be also important that no degradation occurs since soccer players need concurrent abilities of strength, speed, and endurance in the sense of repeated high-intensity actions.

MCT-1 and MCT-4 content in the muscle was not influenced by the intervention of EG. Interventions that showed increases of MCT-1 and MCT-4 conducted higher intensities and metabolic demanding exercises. It is known that high-intensity endurance training increases MCT-1 in trained subjects [41,42] and strength training increases MCT-1 and MCT-4 in untrained subjects [27]. The training program of 3 × 10 maximum jumps and 10 s between each jump was seemingly less intense. Indeed, jumps are metabolically demanding, but 10 s of rest enable adequate delivery of oxygen. Unfortunately, lactate accumulation was not measured during training in the present study. However, hundreds of repeated jumps with 8 s rest between the jumps can result in moderate steady state lactate concentrations of 3–4 mmol·L^−1^ [43]. It might be a question of the stimulus´ intensity or accumulation that need to be analyzed in exercise constellations that increase metabolic stimulus like high intensity interval training. The superimposed WB-EMS on/off-time ratio should be increased accordingly.

Our results show a significant decrease in MCT-4 and MCT-1 content after jump training without EMS (TG), which can be hardly explained, as EG and CG did not show significant differences in MCT-4 and MCT-1 content. Although the EG and TG showed equal training load (see Table 1) generally, high-intensity anaerobic effort in soccer greatly varies according to the playtime and different playing position requirements within a squad [44,45]. Furthermore, there can be differences of intensity in daily soccer training routine that could lead to fluctuations of MCT´s. This speculation is indicated by large standard deviations in total training load of TG (Table 1). Although subjects were assigned to play and train as usual, it was not possible to adjust for this influence in the study. Replication studies with accelerometer-based monitoring of the total loads are required to verify this issue. Additionally, findings warrant further studies about strength training effects to MCT. In this regard, authors have demonstrated the importance of detailed characterization of the training stimulus and the subjects [46,47]. A specification of muscular time under tension and movement dynamics like reactivity are missing in recent studies that dealt with EMS or MCT´s. The reduction effect of MCT could also be attributed to a shift of MCTs to the membrane. A previous study has shown that MCT-1 localisation after training in diabetic patients increased in the sarcolemma of muscle fibers while the sarcoplasmic content was reduced [48].

Some more limitations of the present study have to be mentioned for further research on the effects of WB-EMS to endurance capacities. Due to the small sample size the study has pilot character. Since we did not measure lactate production during training, we are not able to characterize the metabolic stimulus of the training. Moreover, the effects of running performance mainly include improved running economy, time trial performance and sprint performance all of which were not tested in the current work. A more sports-specific testing set such as the Yo-Yo Intermittent Recovery test in combination to sprint tests would have been useful in terms of ecological validity. A Further aspect of limitation is that including players from different teams can result in differences in training sessions for the CG (Table 1). A more detailed documentation of players match and training loads would be helpful to avoid bias. Future research may consider changing study design to evoke higher metabolic stimuli by increasing time under tension, reducing rest intervals or increasing intervention duration. However, the present training stimulus was designed to improve soccer specific high-intensity actions and could be integrated into daily training on a professional level [5].

## 5. Conclusions

We conclude despite findings that the disposed WB-EMS protocol can enhance strength [5] and myofiber adaptations [40] it is not a potent stimulation to improve VO_2_max and lactate transport proteins.

## Figures and Tables

**Figure 1 ijerph-17-01123-f001:**
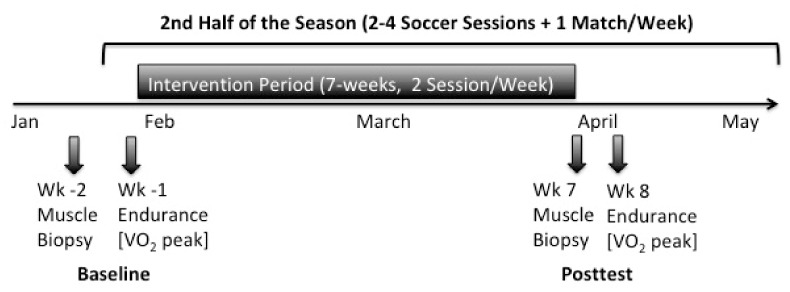
Timeline of endurance testing and muscle biopsy withdrawal during the study in the 2nd half of the season.

**Figure 2 ijerph-17-01123-f002:**
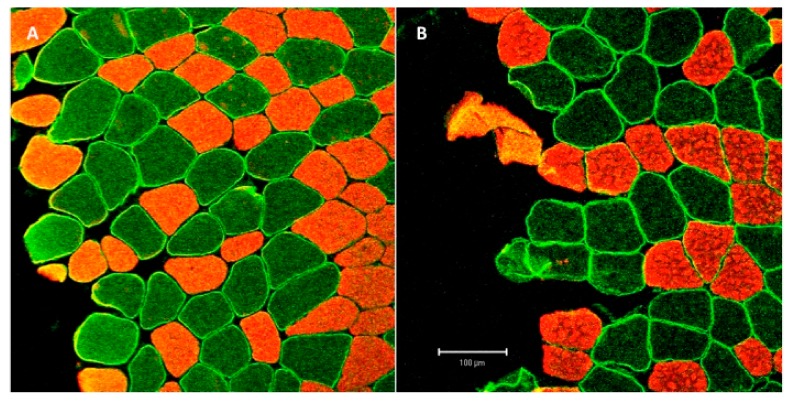
Representative pictures of immunofluorescence stained myofiber cross-sections showing specific MCT-1 and MCT-4 staining (green) and type 1 myofiber staining (red) within membrane and sarcoplasmic areas of myofibers (10× fold magnification). (**A**) MCT-4 Posttest, (**B**) MCT-1 Posttest.

**Figure 3 ijerph-17-01123-f003:**
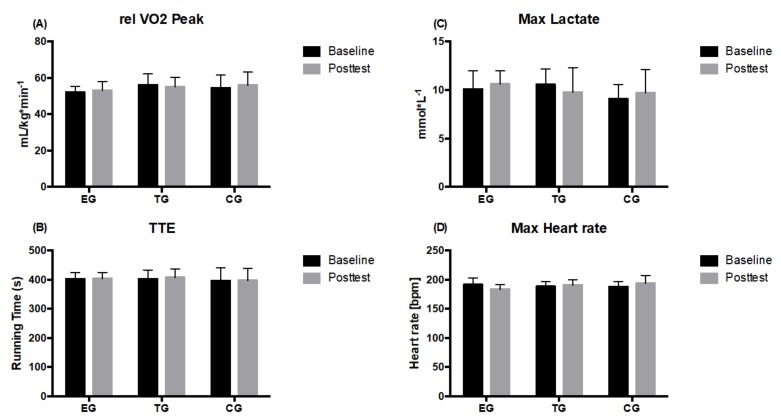
(**A**) Relative maximum oxygen uptake (relVO_2_peak), (**B**) maximal lactate concentration, (**C**) maximal running time till exertion (TTE), and (**D**) maximal heart rate) determined at the endurance ramp-test on the treadmill in EMS-Group (EG), Training-Group (TG) and Control-Group (CG) measured before (Baseline) and after the 7 weeks intervention period (Posttest). Values are presented in means ± SD.

**Figure 4 ijerph-17-01123-f004:**
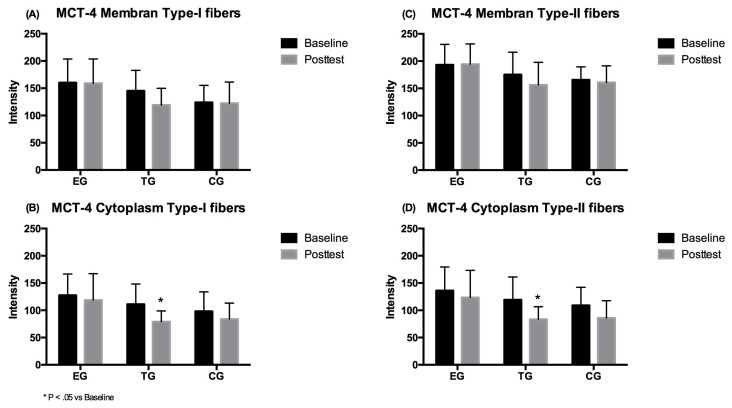
MCT-4 density in type-I fiber (**A**) membrane and (**B**) cytoplasm, and in type-II fiber (**C**) membrane and (**D**) cytoplasm for EMS-Group (EG), Training-Group (TG) and Control-Group (CG) measured before (Baseline) and after the 7 weeks intervention period (Posttest). Values are presented in means ± SD.

**Figure 5 ijerph-17-01123-f005:**
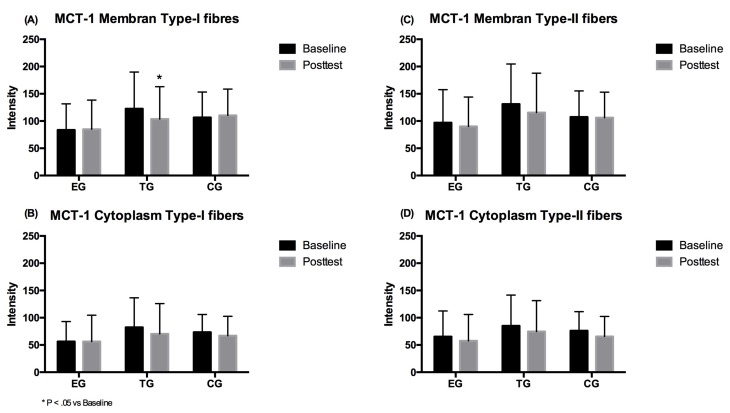
MCT-1 density in type-I fiber (A) membrane and (B) cytoplasm, and in type-II fiber (C) membrane and (D) cytoplasm for EMS-Group (EG), Training-Group (TG) and Control-Group (CG) measured before (Baseline) and after the 7 weeks intervention period (Posttest). Values are presented in means ± SD.

**Table 1 ijerph-17-01123-t001:** Anthropometric data (mean ± SD) and Total Training Load (arbitrary units) during the 7-weeks intervention period calculated by Polar Team-2 Software according to training time spent in defined heart rates zones.

Group	Age [Year]	Height [m]	Weight [kg]	Bodyfat [%]	relVO_2_peak [ml/kg*min^-1^]	Sessions/ Week	Total Training Load [a.u.]
**EG (*n* = 10)**	24.4 ± 4.2	1.82 ± 0.03	81.4 ± 5.3	12.9 ± 2.1	52.1 ± 3.4	3.4 ± 1.2	3431 ± 911
**TG (*n* = 10)**	21.1 ± 1.9	1.83 ± 0.06	79.7 ± 5.5	10.8 ± 2.8	56.3 ± 5.7	3.4 ± 1.3	3479 ± 1723
**CG (*n* = 8)**	23.6 ± 3.9	1.82 ± 0.05	79.7 ± 7.5	14.1 ± 3.6	54.3 ± 7.2	2.6 ± 0.7	2644 ± 1437

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
