# Peer review of "Seven Weeks of Jump Training with Superimposed Whole-Body Electromyostimulation Does Not Affect the Physiological and Cellular Parameters of Endurance Performance in Amateur Soccer Players"

_ijerph, 2020, doi:10.3390/ijerph17031123_

Round 1

Reviewer 1 Report

Dear Authors,

Overall a very well structured research with on-point introduction and well-described methods, results, and conclusions.

There are some small things to answer:

Line 73-74 / Define experience in strength training? how many years was your minimum for inclusion to the study - the same as playing soccer?

Line 73 / They need to play one soccer match per week - define a match - It is not the same load if someone plays a whole match or if it is substituted in the 88th minute. Officially he played a match but from the loading perspective, he shouldn't be in the sample. Report how many minutes of the match did they play and what was your minimum criteria for inclusion in the sample.

Line 74 / were all the players from the same team? If not report from how many. 

Line 77 / Why did you choose only 7 weeks? Explain. The majority of the studies you cite with EMS were longer?

Line 110 / How was the reported Vo2max calculated and when?

In the methods you didn't describe how was body fat, height and weight measured and with what. Add

Kind regards

Author Response

Dear Reviewer#1, thank you very much for your time and efforts revising the manuscript. Language editing has been conducted. Please find our point-to-point with detailed below:

Overall a very well-structured research with on-point introduction and well-described methods, results, and conclusions.

            Reply: Thank you very much for your overall positive impression of the paper.

There are some small things to answer:

Comment 1: Line 73-74 / Define experience in strength training? how many years was your minimum for inclusion to the study - the same as playing soccer?

Reply: Thanks for your constructive and helpful comment. In line with the inquiry of Reviewer#2, all subjects were familiar with different strength training methods, as they have competed on a regional level for the last 3 years with 2-4 training sessions, in part instructed by a strength and conditioning coach. We deleted the sentence of strength training experience and added more comprehensive and detailed information on training volume, experience and contents in the sentence before (line 75).

Comment 2: Line 73 / They need to play one soccer match per week - define a match - It is not the same load if someone plays a whole match or if it is substituted in the 88th minute. Officially he played a match but from the loading perspective, he shouldn't be in the sample. Report how many minutes of the match did they play and what was your minimum criteria for inclusion in the sample.

Reply: We do absolutely agree. In soccer, notable differences in running profiles according to the playing positions and players’ time exists during a match period, but also during training weeks.  That fact contributes to higher inter-individual variability. Unfortunately, we were not able to control for that confounding variable in this study, since the players at this level were not monitored for the percentage proportion of training and game participation. However, covered distance and intensity within the training session has been recorded. Overall, we ensured that players did the predefined training and game loads or did game substitution training with their strength and conditioning coaches. In accordance with your comment, we now addressed this issue within the limitation section (lines 350-352) and now feel that this part improved remarkably.

Comment 3: Line 74 / were all the players from the same team? If not report from how many. 

Reply: Thanks, we added this information to the methods section: Players were from 10 different teams (line 76).

Comment 4: Line 77 / Why did you choose only 7 weeks? Explain. The majority of the studies you cite with EMS were longer?

Reply: Good point – many thanks. We felt that our chosen training volume was sufficiently motivated within the introduction section. IN line with your valuable point, we however added relevant literature that also decided in favor of 7 weeks is sufficient (line 80). Moreover, we felt that a macro-/mesocycle of 6 to 8 weeks suitably fits to training periodization of complementary strategies, such as EMS.

Comment 5: Line 110 / How was the reported Vo2max calculated and when?

Reply: We added this information accordingly: You can now find the details on how to report VO2 in the “endurance test section (lines 146 ff.)” and in the timeline figure (fig. 1). We also included a new and appropriate reference on this issue in the respective “endurance test section (Sperlich et al., 2015)”.

Comment 6: In the methods you didn't describe how was body fat, height and weight measured and with what. Please Add

Reply: Thanks a lot! We added the information for the bioelectrical impedance analysis in line 149, accordingly.

Kind regards

Reply: Thanks again for your valuable inputs and time/efforts spent in order to improve the paper.

Reviewer 2 Report

Title

Please add “amateur” or “regional” before soccer players

Abstract

Lines 14-15: just wondering if the rational should the other way around. I.e., superimposed WB-EMS may positively (or negatively) affect the player’s explosive actions during a soccer game, and hence the player’s performance?

Line 18: “WB-EMS” please don’t use abbreviations in the abstract or explain them.

Introduction

Line 31: not only to tactics and its variability. As the authors mention in the next sentence, the higher distance covered, etc may not only be related to tactics. In this sense I suggest exploring a little this topic, and hence establish a link between the first and second sentence of the intro section.

Line 42: unclear, why? Contradictory findings? Please refer.

Lines 59-60: “limited studies”, and the authors only report one. Please add the main results of the limited studies about this topic.

Line 60: “No study….” So, please add what could be the novelty of this study.

Line 63-65: This sentence “2 short strength 63 training sessions per week could be included in a common training routine in soccer and was sufficient for 64 gains in strength, sprint and jumps [5].” should be deleted from this section.

Line 66: “…may affect…” positively or negatively? During the intro section it could be hard for the readers to understand what is the “good thing”: ideally the superimposed EMS should have a positive or negative effect on the MCT-1 and MCT-4?

Material and Methods

Line 73-74: “Experience in strength training was required.” What does this mean? Strength training was part of the training periodization, was this specific and orientated by a S&C coach? This could be a major issue if the superimposed EMS protocol was not monitored/controlled by a specialist.

Line 75: do you mean an RCT (randomized control trial)?

Table 1: control group present a meaningful lower amount of training load units. Does refers to the specific complement training, right? Does should be explained.

Line 116: please don’t start the sentences with abbreviations.

Line 144: “Endurance test” section lacks of reference(s) to support the protocol.

Line 163: “Muscle biopsies and tissue treatment” section lacks of reference(s) to support the protocol.

Line 171: “Immunohistochemistry” section lacks of reference(s) to support the protocol.

Line 172: why only 26, and not 28?

Line 209-210: Why partial and not eta^2? Please add a qualitative assessment (cut-off agreements) for the effect size.

Results

Whenever using the term “significant” please add the agreement for the effect size (e.g., small, moderate, high).

Lines 217-219: Once again, why the players had different training units due to the complement training. But if you mention that: lines 76-80 “…The EMS group (EG, n=10) performed jumps with superimposed WB-EMS twice a week accompanied by 3 x 10 squat jumps in addition to the daily soccer routine over a period of 7 weeks. To differentiate between the effects caused by EMS and by the squat jumps and soccer training respectively, two control groups were included. A jump training group (TG, n=10) performed the same number of squat jumps without EMS on the same days as the EG.”, the training load of the EG shouldn’t be higher?

Line 220-221: please change this to another place. This does not suit here.

Line 255: non-significant?

Did the authors consider to analyze the data by field position? Players’ intrinsic profile (genetics) could help explaining some outputs…

Discussion

Line 324-325: “hard to explain”…but this is a major “issue”! The protocol (EMS) that could be expected to present a decrease in the MCT counting did not, and the TG did. What are the main differences between the training protocol (TG) and EMS protocol (EG)? Amount of stimulus, recovery time, etc? For example, players in the EG had more time to recover between bouts in comparison to the TG? How this could affect the MCT counting?

Line 324-325: “equal”? Please see previous comment.

Group*Time interaction on MCT-1 type 1 fibers was not discussed…

Time effect on MCT-1 type 2 fibers was not discussed…

If there were no significant differences between groups, why coaches and players should choose to perform a EMS protocol? The standard protocol of the TG as numerous advantages…

Author Response

Dear Reviewer#2, thank you very much for your time and efforts revising the manuscript. Language editing has been conducted. Please find our point-to-point with detailed below:

Title

Comment 1) Please add “amateur” or “regional” before soccer players

Reply: Thanks. We added the word “amateur” in accordance to the text.

Abstract

Comment 2) Lines 14-15: just wondering if the rational should the other way around. I.e., superimposed WB-EMS may positively (or negatively) affect the player’s explosive actions during a soccer game, and hence the player’s performance?

Reply: Thanks for your suggestion that is much appreciated. As we did not determine players explosive actions we are not sure how to address this issue adequately Also, player’s performance during the game was not a main focus. Global physiological read-outs in response to WB-EMS were primary subject of our study. However, we rephrased the wording more towards your recommendation and now ffel that this part gained clarity and readability.

Comment 3) Line 18: “WB-EMS” please don’t use abbreviations in the abstract or explain them.

Reply: Thanks for the eagle-eye. We adapted accordingly.

Introduction

Comment 4) Line 31: not only to tactics and its variability. As the authors mention in the next sentence, the higher distance covered, etc. may not only be related to tactics. In this sense I suggest exploring a little this topic, and hence establish a link between the first and second sentence of the intro section.

Reply: We established a link between the sentences and now feel that it improved readability and comprehensibility. Thank you very much.

Comment 5) Line 42: unclear, why? Contradictory findings? Please refer.

Reply: Good point – thank you. There are no contradictory findings. The reason for the wording was that the authors of the cited WB-EMS study do not explain physiological mechanisms. We deleted the first part of the sentence and feel that it improved notably.

Comment 6) Lines 59-60: “limited studies”, and the authors only report one. Please add the main results of the limited studies about this topic.

Reply: We do absolutely agree. As we could not find further studies which deal with strength training and MCT-expression, we deleted the “limited studies”-sentence and summarized the main results of the strength training study that was published by Juel et al. We attenuated wording at this point, accordingly.

Comment 7) Line 60: “No study….” So, please add what could be the novelty of this study.

Reply: Thanks, we integrated this information.

Comment 8) Line 63-65: This sentence “2 short strength 63 training sessions per week could be included in a common training routine in soccer and was sufficient for 64 gains in strength, sprint and jumps [5].” should be deleted from this section.

Reply: Done.

Comment 9) Line 66: “…may affect…” positively or negatively? During the intro section it could be hard for the readers to understand what is the “good thing”: ideally the superimposed EMS should have a positive or negative effect on the MCT-1 and MCT-4?

Reply: Done. Thank you very much for your constructive comments on the introduction. We feel that our inserted changes improved scientific rigor and clarity.

Material and Methods

Comment 10) Line 73-74: “Experience in strength training was required.” What does this mean? Strength training was part of the training periodization, was this specific and orientated by a S&C coach? This could be a major issue if the superimposed EMS protocol was not monitored/controlled by a specialist.

Reply: In accordance to the comments of reviewer#1, we reworded this part for more clarity and better readability: All subjects were familiar with different strength training methods, as they have competed on a regional level for the last 3 years with 2-4 training sessions per week, in part instructed by a strength and conditioning coach. We deleted the sentence of strength training experience and added more comprehensive and detailed information on training volume, experience and contents in the sentence before (line 75).

Comment 11) Line 75: do you mean an RCT (randomized control trial)?

Reply: Yes, we added this.

Comment 12) Table 1: control group present a meaningful lower amount of training load units. Does refers to the specific complement training, right? Does should be explained.

Reply: Thanks for this hint. We do agree. On this level of play, large variability of training volumes occurs. In line with your comment, we now calculated the standardized mean group differences, adjusted for small sample sizes. Although not statistically significant, we found moderate effects (SMD 0.52, 95% CI: -0.45, 1,46). Moreover, the complement training was not included in the training load. As we decided not to artificially increase soccer-training volume and strength training volume throughout the intervention period, this moderate effects of volume differences could have been expected. In line with your inquiry, we comprehensively point out this important aspect in the limitation section (lines 350-352).

Comment 13) Line 116: please don’t start the sentences with abbreviations.

Reply: We changed the setting of the sentence.

Comment 14) Line 144: “Endurance test” section lacks of reference(s) to support the protocol.

Reply: We added the reference of Sperlich et al. (2015) to support the incremental ramp test protocol.

Comment 15) Line 163: “Muscle biopsies and tissue treatment” section lacks of reference(s) to support the protocol.

Reply: We added the reference of Bergström (1975) to support the muscle biopsies and tissue treatment protocol.

Comment 16) Line 171: “Immunohistochemistry” section lacks of reference(s) to support the protocol.

Reply: We added the reference of Jacko et al. (2019) to support the histology protocol.

Comment 17) Line 172: why only 26, and not 28?

Reply: We had to remove one player from the study because of an ankle joint injury before Post-test. (l. 162). One sample could not be analyzed due to a missing POST test of the subject. This subject was not willing to conduct a second biopsy anymore.

Comment 18) Line 209-210: Why partial and not eta^2? Please add a qualitative assessment (cut-off agreements) for the effect size.

Reply: We added qualitative assessment for the effect size as recommended in Cohen (1988).

Results

Comment 19) Whenever using the term “significant” please add the agreement for the effect size (e.g., small, moderate, high).

Reply: We added the agreement at the appropriate places.

Comment 20) Lines 217-219: Once again, why the players had different training units due to the complement training. But if you mention that: lines 76-80 “…The EMS group (EG, n=10) performed jumps with superimposed WB-EMS twice a week accompanied by 3 x 10 squat jumps in addition to the daily soccer routine over a period of 7 weeks. To differentiate between the effects caused by EMS and by the squat jumps and soccer training respectively, two control groups were included. A jump training group (TG, n=10) performed the same number of squat jumps without EMS on the same days as the EG.”, the training load of the EG shouldn’t be higher?

Reply: You are right. Indeed, there are no significant differences. However, we point out this aspect in the limitation section (lines 350-352).

Comment 21) Line 220-221: please change this to another place. This does not suit here.

Reply: Done. We now placed the information in the training load section.

Comment 22) Line 255: non-significant?

Reply: Thanks again for the eagle-eye. We changed the wording.

Comment 23) Did the authors consider to analyze the data by field position? Players’ intrinsic profile (genetics) could help explaining some outputs…

Reply: You rise an interesting question that we did not address in the study due to the main rationale and sample power. In line with your comment, in soccer great differences in running profiles according to the playing positions exists during match but also during training week what may explain the high inter-individual variability. We cannot control that, even if we would have included only player from one team and programmed every training session. That is the major problem with field studies especially with field sports like soccer. We addressed this point in the limitation section and now feel that this part improved.

Discussion

Comment 24) Line 324-325: “hard to explain”…but this is a major “issue”! The protocol (EMS) that could be expected to present a decrease in the MCT counting did not, and the TG did. What are the main differences between the training protocol (TG) and EMS protocol (EG)? Amount of stimulus, recovery time, etc? For example, players in the EG had more time to recover between bouts in comparison to the TG? How this could affect the MCT counting?

Reply: The study was designed to calculate the effect of EMS, so the focus of our discussion is on the effect on MCTs in EG. However, TG completed the same protocol like EG without superimposed EMS and we see even a negative effect. Days of intervention and recovery time were equal between TG and EG. In line with your comment we do discuss that issue; however it is still difficult to explain and we tried to avoid too much speculation.

Comment 25) Line 324-325: “equal”? Please see previous comment.

Reply: In this sentence we compare the two jump training groups (TG and EG) that had the same total training load (mean: 3430 vs. 3478 and 3.4 sessions/week respectively; see table 1). With respect to lower training volume (not significant; mean: 2644 and 2.6 sessions/week) of CG, please see answer to your previous comment. We point out this aspect in the limitation section (lines 350-52).

Comment 26) Group*Time interaction on MCT-1 type 1 and time effect on MCT-1 type 2 fibers were not discussed…

Reply: It is difficult to discuss these findings because we do not know much about EMS-training or strength training effects to MCTs. However, we addressed that in lines 329-343 and now feel that this part improved.

Comment 27) If there were no significant differences between groups, why coaches and players should choose to perform a EMS protocol? The standard protocol of the TG as numerous advantages…

Reply: In line with your point, based on our findings we now explicitly mentioned that there is no advantage of EMS compared to TG and CG. When aiming to increase anaerobic capacity or using EMS to increase lactate release or uptake in blood or in-between fibers.

Thanks again for valuable and constructive feedback that helped to improve the manuscript.

Round 2

Reviewer 2 Report

The authors made significant improvements, and the manuscript enhanced its quality and can be read clearly.

Just emend some typing errors:

p.e. line 345: change "A Further..." to "A further..."